# Dynamic Species Distribution Modeling Reveals the Pivotal Role of Human-Mediated Long-Distance Dispersal in Plant Invasion

**DOI:** 10.3390/biology11091293

**Published:** 2022-08-30

**Authors:** Christophe Botella, Pierre Bonnet, Cang Hui, Alexis Joly, David M. Richardson

**Affiliations:** 1Centre for Invasion Biology (CIB), Department of Botany & Zoology, Stellenbosch University, Stellenbosch 7602, South Africa; 2Botany and Modeling of Plant Architecture and Vegetation (AMAP), CIRAD, CNRS, INRAE, IRD, University of Montpellier, 34398 Montpellier, France; 3Centre for Invasion Biology, Department of Mathematical Sciences, Stellenbosch University, Stellenbosch 7602, South Africa; 4Biodiversity Informatics Unit, African Institute for Mathematical Sciences, Cape Town 7945, South Africa; 5Inria, LIRMM, University of Montpellier, 34095 Montpellier, France; 6Department of Invasion Ecology, Institute of Botany, The Czech Academy of Sciences, 252 43 Průhonice, Czech Republic

**Keywords:** biological invasions, population dynamics, long-distance dispersal, human-mediated dispersal, matrix population models, Bayesian inference, species distribution models, citizen science, presence-only data

## Abstract

**Simple Summary:**

Understanding biological invasion mechanisms is crucial to design effective management strategies preventing their impacts on ecosystems. If the role of long-distance dispersal and age-dependent fecundity in plant invasion speed has been characterized in theory, empirical support is still rare given the difficulty to jointly fit demographic and spread parameters of dynamic models to available data. We proposed a statistical model to fit such parameters to heterogeneous observations collected across space and time. We can directly test hypotheses on the importance of various mechanisms in a past invasion. We demonstrated the potential of this method by determining the roles of human-mediated long-distance dispersal and age-dependent fecundity in the invasion of the shrub *Plectranthus barbatus* in South Africa. Our model revealed a massive wave of spread driven by human-mediated long-distance dispersal and originating from the cities of first introduction. The species completed its invasion of all favorable areas a few years after, in the mid-1990s. Without human-mediated long-distance dispersal, the maximum population would have been obly 30% of the current population. The delayed reproductive maturity explained the invasion lag phase. It highlights the importance of early eradication of weedy horticultural alien plants around urban areas to hamper the invasive spread.

**Abstract:**

Plant invasions generate massive ecological and economic costs worldwide. Predicting their spatial dynamics is crucial to the design of effective management strategies and the prevention of invasions. Earlier studies highlighted the crucial role of long-distance dispersal in explaining the speed of many invasions. In addition, invasion speed depends highly on the duration of its lag phase, which may depend on the scaling of fecundity with age, especially for woody plants, even though empirical proof is still rare. Bayesian dynamic species distribution models enable the fitting of process-based models to partial and heterogeneous observations using a state-space modeling approach, thus offering a tool to test such hypotheses on past invasions over large spatial scales. We use such a model to explore the roles of long-distance dispersal and age-structured fecundity in the transient invasion dynamics of *Plectranthus barbatus,* a woody plant invader in South Africa. Our lattice-based model accounts for both short and human-mediated long-distance dispersal, as well as age-structured fecundity. We fitted our model on opportunistic occurrences, accounting for the spatio-temporal variations of the sampling effort and the variable detection rates across datasets. The Bayesian framework enables us to integrate a priori knowledge on demographic parameters and control identifiability issues. The model revealed a massive wave of spatial spread driven by human-mediated long-distance dispersal during the first decade and a subsequent drastic population growth, leading to a global equilibrium in the mid-1990s. Without long-distance dispersal, the maximum population would have been equivalent to 30% of the current equilibrium population. We further identified the reproductive maturity at three years old, which contributed to the lag phase before the final wave of population growth. Our results highlighted the importance of the early eradication of weedy horticultural alien plants around urban areas to hamper and delay the invasive spread.

## 1. Introduction

Biological invasions are a major driver of the current biodiversity crisis and incur tremendous cost to human societies [1]. Many studies have quantified the massive and growing socio-economic costs of invasions. A recent estimation of the overall cost of invasive species at the European scale quantified the amount to be approximately USD 140 billion [2] between 1960 and 2020, including both management (40%) and damage (60%) costs. Although country-level assessments identified heterogeneous costs and knowledge gaps [3,4,5] all the studies report a clear rise in the costs and suggested that these reported costs were severely underestimated due to the lack of data for many taxa. The costs to humans and biodiversity will continue to grow given the trend of ongoing invasions [6] and the inertia of the invasion processes due to invasion debt [7] highlighting the need to identify early these introduced species that are likely to become invasive and cause impacts. Indeed, better estimation of the residence times and elucidation of the causes of lag phases [8] are crucial for improving the efficiency of the management [9]. However, efforts directed at early detection often incur costs and risks without an obvious return on investment, hence the recent drive for crowdsourcing to overcome data [10].

Effective management involving the early detection and prevention of the negative impacts of plant invasions demands a robust understanding of the past transient spatial dynamics on which to base predictive scenarios of future invasions. It is crucial to acknowledge the non-equilibrium reality of an introduced species with its invaded environment due to its limited residence time and dispersal constraints (i.e., the non-equilibrial invasion dynamics)[11], which must be accounted for when fitting invasive species distribution models [12], especially given the ongoing niche shifts under climate change [13]. It is also necessary to account for the fact that the native range of species is often restricted when compared to their potential environmental range [14]. Quantitative calibration of dispersal kernels and demographic rates to reconstruct invasion dynamics has often been based on literature reviews [15,16], while estimates of such parameters in controlled experiments have revealed high intrinsic variability across repetitions, even under identical conditions [17].Although such estimates are useful, they lack coherence with historical invasions. Statistical approaches have been used to circumvent the problem by highlighting the relationship between invasiveness and the biological traits related to reproduction and dispersal ability [18] or by identifying invasive-prone phylogenetic groups [19,20]. As a complementary approach to allow for more consistent interpretations with evidence from past invasions, we advocate for fitting directly mechanistic parameters from spatio-temporal observational data. This approach, also called mechanistic-statistical modeling, has the potential to combine the strengths of the so-called correlative and process-based species distribution models [21]. It was recently used to jointly fit demographic and spread rates with environmental responses from large-scale observational data in modeling the recolonization of wolves [22] and the invasive spread of the watermelon mosaic virus [23]. However, it has never been used to reconstruct perennial plant invasions as far as we know. Bayesian inference is an attractive framework for this purpose as it allows us to account for the joint uncertainty of the environmental suitability, spread, and demographic parameters, and to explore all the possible combinations that best explain the data. It also allows us to define a level of constraint on each parameter around realistic values using prior knowledge.

To design a sound dynamic model of invasion, we must ensure the coherence of the model with plant physiology and spread mechanisms while simultaneously avoiding over-parameterization for relevant and reliable interpretations. In particular, if environmentally suitable areas are prerequisites for invasion success, the ability to produce propagules at an early age and in sufficient numbers can further enhance invasion success [24]. Several studies have shown that natural short-distance dispersal cannot explain the fast rate of invasive spread and have highlighted the crucial role of long-distance dispersal [25,26]. More importantly, invasion success is often strongly related to the intensity and duration of the human use of the invading species [1,27,28] highlighting the importance of introduction pressure and human-mediated dissemination. For plants, the proximity to urbanized areas and road network hubs favors the long-distance dispersal of seeds through the movement of plants and soil. The variation of the fecundity across life-stages and their relative abundances in a population are another important driver of the speed behind invasion waves [26]. The so-called population projection matrix (PPM) models have been developed to account for these factors [29,30]. Such stage-dependent demographic rates are particularly influential in driving the invasion dynamics of woody plants, especially given the variation of maturation age and the large increase in fecundity with age [31]. All these factors must be captured when designing a dynamic model for explaining the high variability of lag phases [32,33,34] and the eventual success of plant invasions.

We propose a Bayesian dynamic species distribution model to investigate the role of human-mediated long-distance dispersal and age-structured fecundity in the large-scale spatial dynamics of invasive perennial plants. This model is similar to other Bayesian dynamic models [22,23,35,36] but differs in the ecological processes it models and the type of data it uses. In a nutshell, our dynamic state-space model [37] is decomposed into two main components: the ecological process model, determining the hidden population dynamics, and the sampling process model, determining the likelihood of the observations given the hidden population states. The ecological process model is itself decomposed into a model of the initial population states and a Markov model of the transition between years. Given some parameters, the latter computes deterministically the temporal succession of the hidden age-structured populations over a spatial lattice of sites. We account for human-mediated long-distance dispersal by including a gravity dispersal kernel where the portion of seeds dispersed over long distances is proportional to the percentage of urban land cover in a lattice cell. We also explicitly model age-structured fecundity and its demographic effect using a type of population projection matrix (PPM) model [30]. The sampling process model computes the likelihood of the spatio-temporal presence-only records given the hidden population states, accounting for spatio-temporal variations in the sampling effort and for variable detection rates across data sources. Such a data integration approach [38] allows us to capitalize on the availability of large numbers of recent records from large-scale crowdsourcing platforms (e.g., iNaturalist, Pl@ntNet) and older records from herbaria and institutional monitoring schemes to reconstruct the spatial dynamics of invasive plant species over several decades. The model is fitted in a Bayesian framework, allowing the use of a priori knowledge through informative prior demographic parameter distributions, in order to gain information on the detection rates and, hence, to address parameter identifiability issues between the sampling effort and abundance [39]. 

We address, as a demonstration, the invasion of *Plectranthus barbatus* in the Southern Cape region of South Africa as a case study. The species is native to tropical India and likely to northeast Africa [40] was most likely introduced to South Africa around the middle of the 20th century. We could identify the likely foci of introduction areas using herbarium records. It was classified as invasive in South Africa in 2004 (Category 1b taxon in the NEM:BA regulations) but has never been targeted in any substantial control program. In the following, we introduce our model, the study area, the data, and the model validation method. We estimate the model parameters to reconstruct the spatial dynamics of *Plectranthus barbatus* and investigate the role played by human-mediated long-distance dispersal and age-structured fecundity. We show maps of the relative population and of the invasion syndrome across cells for the key years of the invasion. We also compare the actual trajectory with modeled alternative trajectories where the model is deprived of long- or short-distance dispersal to better assess their relative importance to the invasion dynamics.

## 2. Materials and Methods

### 2.1. Ecological Process Model

#### 2.1.1. General Structure

We propose a lattice-based spatio-temporal model with discrete time steps (years) to represent the spatial dynamics of the age-structured population of a perennial plant species. Each year, a number of seeds are produced by each plant depending on its age. Spatial seed dispersal is structured with the same framework as in [16]; specifically, the seeds are partly dispersed locally in the cell, to adjacent cells (short-distance dispersal), and to any other cell in a proportion that is assumed to depend on the local urbanization rate (human-mediated long-distance dispersal). Then, for a given cell, the proportion of seeds that grow into a new plant depends on the local carrying capacity (self-regulation) and the environmental suitability. The state of the population in cell *i* (=1, 2, ..., *C*) during year t is divided into the population sizes per age $\left(n_{i,t,1},\ldots ,n_{i,t,K}\right)$ up to *K* = 50 years, assumed to be the maximal age for *Plectranthus barbatus,* which is optimistic for this shrub. This model is thus a matrix population model, a class of well-studied demographic models [29] but rarely calibrated on real data. We express the transition rule, i.e., the population sizes in year *t + 1* based on their states in the previous year, in Equation (1) for plants older than one year and Equation (2) for the newly established plants.
(1)$$\forall k>1,n_{i,t+1,k}=round\left(\left(1-\rho \right)n_{i,t,k-1}\right)$$

In Equation (1), the parameter ρ is the mortality, i.e., the death probability of any plant from one year to the next, which may also include removal by humans (e.g., management) or other animals. We inform its prior distribution using the maximal age *K* of a plant and the maximal carrying capacity, as explained in Section A.2. The function *round* returns the closest integer which prohibits seed production from less than one adult plant and thus allows the possibility of total extinction of a local population, e.g., following temporal changes of the environment. The mortality is assumed to be independent of plant age, which leads to a geometrically decreasing probability of survival with advancing age.
(2)$$n_{i,t+1,1}=round\left(s_{i,t}c_{i,t}p_{i,t+1}\right)$$

The above Equation (2) depicts the number of newly established plants (of age 1) at year *t +* 1 in cell *i*, which is the product of the number of seeds $s_{i,t}$, the proportion of seeds surviving self-regulation $s_{i,t}$, and the environmental suitability during the new year $p_{i,t+1}$. Each term is expressed as follows:


-

$s_{i,t}=\sum_{i^{\prime }=1}^{C}disp\left(i,i^{\prime }\right)\sum_{k^{\prime }=1}^{K}f_{\gamma }\left(k^{\prime }\right)n_{i^{\prime },t,k^{\prime }}$




is the number of seeds that have spread to cell i at year t (propagule pressure) which might establish and become new plants at *t +* 1, and $f_{\gamma }\left(k\right)$ is the fecundity of any plant of age *k*, i.e., the number of seeds produced by a plant in one year, parameterized with γ (see Equation (3)).


-$disp\left(i,i^{\prime }\right)=1/D_{i^{\prime }}$ if $i=i^{\prime }$ (within-cell dispersal),


or $d_{s}/D_{i^{\prime }}$ if $i^{\prime }\in Neig\left(i\right)$ (adjacent-cell short-distance dispersal),

where $Neig\left(i\right)$ is the set of neighbor cells of cell *i*, including 3 to 8 cells for our study area,

or $d_{l}a_{i^{\prime }}/D_{i^{\prime }}$ otherwise (long-distance dispersal)

is the portion of seeds produced in *i*’ which is dispersed to *i* so that $\sum_{i}disp\left(i,i^{\prime }\right)=1$, where the normalization constant is defined as follows
$$D_{i^{\prime }}=\left(\overset{C}{\underset{i^{\prime \prime }=1}{\sum }}1_{i^{\prime \prime }\neq i^{\prime },i^{\prime \prime }\notin N\left(i^{\prime }\right)}\right)d_{l}a_{i^{\prime }}+\left(\overset{C}{\underset{i^{\prime \prime }=1}{\sum }}1_{i^{\prime \prime }\in N\left(i^{\prime }\right)}\right)d_{s}+1$$

Hence, $disp\left(i,i^{\prime }\right)$ depends on two parameters:-$d_{s}>0$ determines the proportion of seeds from *i*’ that reach *i* if the latter is adjacent (short-distance dispersal), noted $i^{\prime }\in N\left(i\right)$-$d_{l}a_{i^{\prime }}$, where $a_{i^{\prime }}$ is the proportion of urban habitat area in cell *i*’, determines the proportion of seeds from *i*’ transported via long-distance dispersal to *i*, i.e., when *i* is not adjacent to *i*’, noted $i^{\prime }\neq i,i^{\prime }\notin N\left(i\right)$. In other words, long-distance dispersal diffuses a portion of seeds homogeneously and instantaneously across the domain.-The proportion of seeds from *i* participating in local recruitment is set as the reference in this parametrization.

Note that all seeds produced are always distributed among the domain cells following the formulation above, even for a border cell. It induced a negligible difference of spread behavior between border and core cells due to the extremely low fitted values for $d_{s}$ and $d_{l}$ in our study.

$$c_{i,t}:=max\left(\frac{p_{i,t+1}A_{i}\varphi -\sum_{k=2}^{K}n_{i,t+1,k}}{max\left(p_{i,t+1}A_{i}\varphi ,s_{i,t}\right)},0\right)\in \left[0,1\right]$$
is the ratio of seeds surviving self-regulation, including competition with existing plants. This ratio is proportional to the difference between the cell’s carrying capacity ($p_{i,t+1}A_{i}\varphi$, i.e., the maximum number of plants the cell can host) and the local population size of age 2 or older in the next year ($\sum_{k=2}^{K}n_{i,t+1,k}$). Parameter φ is the maximum number of plants in a fully terrestrial cell (whose prior distribution uses external information from [16] see Section A.2); $A_{i}$ is the proportion of land; and $p_{i,t+1}$ modulates the carrying capacity given the environmental suitability of the cell (see below). This formulation assumes that the cell will have $p_{i,t+1}A_{i}\varphi$ total slots in the next year, from which we remove the number of slots occupied by the plants surviving until the next year in order to obtain the number of available slots for seeds to potentially grow into new plants. Each incident seed is then distributed to one available slot, and all extra seeds are discarded. When the number of incident seeds $s_{i,t}$ gets larger than the cell’s carrying capacity $p_{i,t+1}A_{i}\varphi$, the denominator $max\left(p_{i,t+1}A_{i}\varphi ,s_{i,t}\right)$ ensures that the number of plants will not exceed the carrying capacity in the following year.
$$p_{i,t}:=p_{\beta }\left(x_{i,t}\right)=exp\left(\beta^{T}x_{i,t}\right)/\left(1+exp\left(\beta^{T}x_{i,t}\right)\right)\in \left[0,1\right]$$
parameterized by β, models the probability that any seed that has not been discarded due to self-regulation produces a new plant of age 1 given the environmental variables in the cell at the current year $x_{i,t}$. As it also modulates the carrying capacity, $p_{i,t}$ measures the environmental suitability. We assume here that a seed either produces a plant or dies in the year, and we do not account for seed bank dynamics.

Assuming a constant environment and that the propagule pressure exceeds the cell’s carrying capacity, the population increases by approximately $\varphi^{\prime }\left(p+\left(p\rho -p-\rho \right)\left(\sum_{k=1}^{K-1}n_{i,t,k}\right)/\varphi^{\prime }\right)$, where $p:=p_{i,t+1}$ $\varphi^{\prime }:=p_{i,t+1}A_{i}\varphi$, to simplify the notations. Hence, as long as the population is negligible compared to the carrying capacity, the former converges towards the latter at geometric speed (the gap to the carrying capacity is decreased by a ratio p between *t* and *t +* 1). The mortality hampers the growth more as the population grows.

#### 2.1.2. Age-Structured Fecundity

The number of seeds produced by a plant in a year, i.e., its fecundity, is assumed to be zero during the first years, as the plant needs to reach a critical photosynthetic ability that enables reproductive maturity [31]. In a second phase, fecundity is assumed to scale to a power of the diameter. However, [31] demonstrated that this property, called allometric scaling, rarely holds for large plants, which most often exhibit fecundity saturation. Fecundity saturation may affect our data given our study time interval exceeding 40 years. In the absence of a mechanistic model to explain this phenomenon [31], we formulate fecundity as a function of plant age integrating maturity, allometric scaling, and saturation, as given in Equation (3).
(3)$$f_{\gamma }\left(k\right)=floor\left(\frac{Mk^{\theta }}{M{\hat{k}}^{\theta }+k^{\theta }}\right)$$

In Equation (3), $\hat{k}$ is the age of reproductive maturity; θ is the allometric scaling factor controlling the fecundity growth rate once maturity is reached; and M is the maximum fecundity (number of seeds per plant). The floor function ensures that the fecundity is zero until the first year of seed production $\hat{k}$. This parameterization of $f_{\gamma }\left(k\right)$ generates a sigmoid-type curve with interpretable parameters.

#### 2.1.3. Initial Populations 

A last component of the model is the initial population state in the first year. Most naturalized plant species were introduced many years before the first naturalized population was established. This complicates the modeling of the invasion dynamics back to the very first introduction. We assumed that, in the first model year (1980), *Plectranthus barbatus* was still restricted to the cells where at least one record was reported before 1980, thanks to the BODATSA database. The species was initially reported in $N^{ini}=24$ cells noted $i_{1}^{ini},\ldots ,i_{N^{ini}}^{ini}$, and we assumed it to be absent from all other cells in this first year. The fact that these 24 cells were in large urban areas supports the hypothesis that the species was still mostly growing as cultivated specimens by then and, hence, our assumption that it was absent from wilder areas. We accounted for the uncertainty pertaining to the initial population size and age structure in each initial presence cell *m* with two parameters: $popIni_{m}$ and $ageRatio_{m}$. More precisely, we set $n_{i_{m}^{ini},1,k}=popIni_{m}{\left(ageRatio_{m}\right)}^{k}{\left(1-ageRatio_{m}\right)}^{K-k}K!/\left(k!\left(K-k\right)!\right)$, i.e., the age-structure of the population is deterministic and proportional to the probability function of a binomial distribution of the parameters $ageRatio_{m}$ and *K* (maximal plant age set as 50), so that $K\times ageRatio_{m}$ gives us the mean age of this population, while the total cell population size is given by the parameter $popIni_{m}$. 

### 2.2. Sampling Process Model

For each dataset *d*, the number of records observed in cell **i** during year *t* is modeled with a Poisson distribution:(4)$$y_{i,t,d}∼P\left(n_{i,t,.}o_{i,t,d}\right)$$
where $o_{i,t,d}:=p_{d}^{detec}N_{i,t,d}^{TG}$ is the sampling effort on the focal species located in cell *i* and at year *t* from dataset/monitoring scheme *d* and where $n_{i,t,.}=\sum_{k=1}^{K}n_{i,t,k}$ is the total plant population of cell *i* at year *t*. $N_{i,t,d}^{TG}$ is the total number of records for the target group (TG) of the species [41] specifically chosen for dataset *d* as a proxy of its sampling effort intensity across space and time, as explained further. The parameter $p_{d}^{detec}$ is meant to account for the global variability of the reporting interest across the datasets. This model implicitly assumes that all plants of a cell for a given year are independently sampled with the same small probability ($o_{i,t,d}\ll 1$). Indeed, under the assumption that $o_{i,t,d}\ll 1$ and if $n_{i,t,.}$ is big (≳30), the binomial distribution of $y_{i,t,d}$ of the parameters $o_{i,t,d}$ and $n_{i,t,.}$ is well approximated by the Poisson distribution given above. 

To select a TG for each dataset, we needed to account for the spatial distribution of its constituent species, as it could induce a strong bias in our estimation of the species distribution parameters. Specifically, we would ideally need the underlying sum of the TG species abundances to be as homogeneous in space as possible [42] As a simple heuristic to approach this situation and minimize bias, we included the species in the TG one at a time, beginning with the focal species, such that, at each step, the added species maximized the spatio-temporal volume occupied by the TG globally. More precisely, at each step, the newly selected species is the one that adds the most spatio-temporal volumes, i.e., combinations of spatial cell and year that are absent from the current TG at this step. If no species adds any spatio-temporal volume, we take the species that maximizes the Shannon entropy of the distribution of TG species richnesses across spatio-temporal volumes. The procedure stops when all the species are added to the TG or if all the remaining species decrease the value of this entropy. The initial species included in the TG is the focal one, in order to ensure that $y_{i,t,d}=0$ when $N_{i,t,d}^{TG}=0$. This simple procedure favors a TG whose records cover the largest area while controlling for the spatial heterogeneity in richness and, hence, avoiding combinations of species with correlated spatial distributions.

### 2.3. Data

#### 2.3.1. Temporal and Spatial Extent

Our study area is a rectangle including most of the Western Cape and the western end of the Eastern Cape province of South Africa (Figure 1). The study area comprises 817 square 5 min terrestrial cells (approximately 10 km size), a scale defined based on the resolution of the past occurrence data described in the next paragraph. The temporal extent of our model is 42 years (1980 to 2021). 

#### 2.3.2. Occurrence Data

We used the presence-only records from the three largest plant occurrence datasets in South Africa with distinct aims and complementary temporal coverage: the BODATSA database (GBIF extraction on the 01/07/21: https://doi.org/10.15468/dl.9q36sf, accessed on 20 August 2022) gathers the records from the professional botanists of South African herbariums since the early 20th century and was crucial to assessing the pioneer locations of *Plectranthus barbatus* in 1980; SAPIA is a professional monitoring scheme that collates records of alien plant species and was especially active between 1988 and the early 2000s; and iNaturalist is a naturalist community platform which has collected a large amount of validated (research-grade) crowdsourcing records in South Africa since 2017. However, the records from BODATSA and significant parts of those from SAPIA are rasterized over spatial grids. We dealt with this spatial uncertainty by aggregating the records as counts per dataset, year, and cell of a common grid of 5 min square cells (approx. 10 km size), being the one used by many SAPIA records. We thus build the three-dimensional array of focal species records noted y and the one of TG records noted $N^{TG}$. The BODATSA records are rasterized over a quarter-degree grid, which is too rough for our analysis, and each cell of this grid exactly overlaps 9 cells of our 5 min grid. Indeed, our grid is nested inside the quarter-degree grid. Thus, we split each BODATSA record into 9 records, each one being associated to one 5 min cell overlapped by the quarter-degree cell and being given the same weight of 0.1111, overall summing to 1. In mathematical terms, this weight corresponds to the probability that the record belongs to a particular 5 min cell, assuming that its unknown location is randomly distributed across the original quarter-degree cell area, and aims at accounting for the spatial uncertainty associated with this record. For iNaturalist, we only used research-grade records, i.e., those that have had at least one external review by a community member and whose species final identification has at least a two-thirds agreement among reviewers; these were extracted on 12/02/21. Our data comprised 190 total records for *Plectranthus barbatus.* The automatic selection of TG plant species per dataset led to the selecting of 354, 169, and 348 species, respectively, for BODATSA, SAPIA, and iNaturalist, representing a total of 106,287 TG records collected during 1980 and 2021. 

#### 2.3.3. Environmental Variables

We used spatio-temporal environmental variables in two components of the proposed model: the germination rate $p_{\beta }\left(x_{i,t}\right)$ that we formulated as a function of vector $x_{i,t}$, including land cover and bioclimatic variables, and the long-distance dispersal rate, which depends on the urban percent cover $a_{i}$ of a cell *i*.

**Land cover variables**. We focused on percent-cover variables for summarizing the land cover for each cell, i.e., the percentages of the land covered by forests, crops, and settlements, respectively, in each cell. The latter is used for modeling $p_{\beta }$ and the (long-distance) dispersal kernel $disp\left(.,.\right)$. The land cover has changed substantially in recent decades, affecting the spatial invasion dynamics. To account for these changes in our model, we reconstructed them by linearly interpolating our percent-cover variables between the 4 sampling years where global coverage datasets were available, namely 1992, 2001, 2010, and 2019. We also linearly extrapolated outside of these sampling years. For the year 1992, we used GLCC-IGBP [43] (accessed on 15 October 2021) with IGBP land cover classification. For the other years, we used MODIS/Terra+Aqua Land Cover Type Yearly L3 Global 500 m SIN Grid (accessed on 15 October 2021), which follows the same land cover classification.

**Terrestrial area within cells.** We computed each cell’s terrestrial area based on the sum of the terrestrial land cover areas for that cell using MODIS 2019 (our most recent land cover raster) and considered it as a constant over the study period.

**Bioclimatic variables.** We collated historical monthly weather data from CRU-TS 4.03 [44], downscaled with WorldClim 2.1 [45] and accessed through: https://worldclim.org/data/monthlywth.html (accessed on 20 August 2022). This dataset provides a worldwide coverage at a 2.5-arc-minute resolution of monthly maximum/minimum temperature and precipitation. We derived from it (i) the annual mean diurnal range (mean of monthly max–min temperature); (ii) the maximum temperature of the hottest month (max. of monthly mean of daily max. temperature); (iii) the minimum temperature of the coldest month (min. of monthly mean of daily min. temperature); (iv) the precipitation (v) the precipitation of the wettest month; and (vi) the precipitation of the driest month for each year from 1980 to 2018. Indeed, these six bioclimatic variables have been used to model the invasive species’ distribution response to climate change [46]. We extrapolated the values in each 2.5-arc-minute cell for the years from 2019 to 2021 based on the prediction of a simple linear regression fitted over the years from 2000 to 2018. We upscaled each raster to our 5 min grid scale by averaging over the 2.5-arc-minute cells whose center fell inside each 5 min cell. Finally, to restrict the number of parameters, we centered and scaled each of all of the six variables and synthesized them by taking the values along the two first axes of a singular value decomposition (SVD) of the bioclimatic variable values across the combinations of cells and years. This pipeline is provided in our GitHub script repository: https://github.com/ChrisBotella/plectranthus_barbatus (accessed on 20 August 2022).

### 2.4. Model Fitting, Convergence Assessment and Posterior Samples

We used prior knowledge from the literature and our own insights on invasive shrub ecology to constrain the parameter prior distributions and improve parameter estimability (See Section A.2). We fitted the model parameters on our data using a Monte Carlo Markov chain (MCMC) algorithm with Metropolis–Hastings sampling (Figure 2). We used our own implementation to better control the cost in memory. Convergence of the MCMC was difficult to obtain given the complex and discontinuous surface of the unnormalized posterior log-likelihood (hereafter called posterior likelihood), and in such cases, it is much more efficient to start the MCMC in the region of the parameter space with maximal posterior likelihood. We ran three successive sessions of nine independent chains for 100,000 iterations to progressively refine the initialization parameters $\Theta_{0}$. While $\Theta_{0}$ was randomly drawn based on prior distributions independently for each chain for the first session, it was set as the parameters maximizing the posterior likelihood across all samples from the past session for the second and third sessions. Box 1 summarizes the model fitting procedure. To assess the convergence of the last MCMC session, we analyzed visually the parameter trace plots, including all chains (Figure A2 of Section A.3), and computed their univariate and multivariate Gelman and Rubin criteria (Brooks & Gelman, 1998), implemented in the R package **coda** (see Section A.3). We used 1678 posterior samples in the main analysis, obtained from a thinning of samples from the last MCMC session after applying a burnin of 15,000 iterations. We set a thinning interval of 450 to reduce the auto-correlation between MCMC samples due to a very low acceptance rate. 

### 2.5. Output Representations

**Model interpretation.** Given the lack of identifiability of certain parameters, we had to be careful when interpreting the model output. For example, it is nearly impossible to disentangle population size from the reporting interest $p_{d}^{detec}$, due to our presence-only data (see a discussion about the problem in Hastie & Fithian, 2013); so, their identifiability mostly relied on the constraints imposed by their prior distribution (see Section A.4 for comments on parameter correlations and interpretation), and such parameters had a large posterior confidence interval. Consequently, our interpretations did not rely on the absolute estimated population size but compared its value or order of magnitude between spatial cells and/or from one year to another (relative population size).

**Population size percentile spatio-temporal maps.** We represented the relative population size across cells and years to understand the species’ invasion dynamics. For this purpose, we computed, for each parameter sample, the population size per couple (cell, year) and discretized it by associating each to one of the percentile intervals [0, 40th percentile], ]40th percentile, 70th percentile], ]70th percentile, 90th percentile], and ]90th percentile, maximum]. Based on this discretization scheme, we propose a simple way to deal with ambiguous couples (cell, year) for which there is no consensus across parameter samples: Each couple where less than two-thirds of the samples agree with their population size percentile is tagged as “uncertain”. We were thus able to draw comparable maps of relative population size for selected years in Figure 3 and for all years in our interactive web Section A.6 (R Shiny application): https://chrisbotella.shinyapps.io/plectranthus_barbatus_sa_maps/ (accessed on 20 August 2022)

**Invasion syndrome spatio-temporal maps.** We analyzed the growth and spread status of populations across space and time. We simplified the growth and spread status of a population by defining five discrete categories, called invasion syndromes: (i) certain population growth but uncertain dispersal (the population does not spread enough seeds for new plants to effectively grow from it in other cells); or (ii) certain population growth and dispersal (e.g., invasive population); or (iii) certain dispersal but no certainty about whether the population is growing or declining (e.g., large invasive population which is self-regulated); or (iv) certain population decline and dispersal (e.g., a large population declining due to environmental change); or (v) certain population decline but uncertain dispersal (e.g., a collapsing population). The growth status and spread status of any population are determined exactly and independently for each posterior sample. The growth (resp. decline/dispersal) status of a population is considered certain if at least two-thirds of the posterior samples agree that the population is growing (resp. decreasing/generating new plants in other cells) from one year to the next. If the status of growth and dispersal does not belong to any of the five invasion syndromes, we tag the population as uncertain. We map the invasion syndrome across the cells for 6 key years in Figure 4 and for all the years in our interactive web Section A.6 (R Shiny application): https://chrisbotella.shinyapps.io/plectranthus_barbatus_sa_maps/ (accessed on 20 August 2022). We also reconstructed the total population, seed production, shanon entropy of population across cells per year (Figure 5) and the number of new trees disseminated by long-distance dispersal per year (Figure 6).

**Simulated trajectories under restricted dispersal modes.** We compared the relative importance of short-distance dispersal and human-mediated long-distance dispersal in the past invasion dynamics by carrying out an ablation simulation experiment. We simulated the past invasion dynamics again for each posterior sample by removing either (i) the long-distance human-mediated dispersal (setting $d_{l}=0$) or (ii) the short-distance dispersal (setting $d_{s}=0$) in the model, with all the parameters otherwise unchanged. For each posterior sample, we computed yearly the difference between the ablated and the full model for the population size and the Shannon entropy of population sizes across cells and divided it by the full model value. We show the average across all the posterior samples (solid curve) and the 90% confidence interval (ribbon) of this “relative difference” for the population size (top) and the Shannon entropy (bottom) in Figure 7. Note that the first scenario, where we removed the long-distance dispersal, is closely related to the scenario where the plant would be systematically eradicated from every cell having a non-null urban area. The latter would have an even more drastic negative impact because the plants in the urban cells do not even contribute to the local and short-distance dispersal.

**Fecundity versus plant age.** Exploring the estimated fecundity, i.e., the number of seeds produced as a function of plant age, as defined in Equation (3), enables us to test our hypothesis on the role of the delay before reproductive maturity in determining the invasion dynamics. Figure 8 shows the posterior distribution of the fecundity as a function of plant age divided by its asymptote (M). 

## 3. Results

We analyzed the model parameters’ estimability and the predictive performance in time. We produced a map-based reconstruction of the key years of *Plectranthus barbatus’* invasion history in the Southern Cape region of South Africa in Figure 3 and Figure 4. We also explored several questions pertaining to the role of human-mediated long-distance dispersal and age-structured fecundity in this invasion event.

**A marked information gain on most parameters despite imperfect MCMC convergence.** Despite the sampling heterogeneity of our presence-only data, the convergence of the MCMC chains was reasonable for most parameters from the visual inspection of their trace plots and univariate convergence criteria (PSRF < 2), but the multivariate criterion (MPSRF = 2.8) suggested that the algorithm did not fully converge to the posterior distribution (see Section A.3). This can be partly explained by a strong negative correlation between the detection rates and the maximal carrying capacity (Figure A7 of Section A.4). This suggests that the strong observed information gain on the detection rates (Figure A4 of Section A.4) came mainly from the informative prior distribution of the carrying capacity (see Section A.2). There was also a gain of information on mortality, with a posterior mean of 0.5, even though this parameter did not converge well, showing a large 95% confidence interval of 0.36. Most other parameters, including the initial population parameters, visually showed a strong information gain (Figure A4, Figure A5 and Figure A6 of Section A.4), except the maximal fecundity M, whose posterior sample distribution was very similar to the prior distribution. 

**Model validation.** The model predictions in the validation depended on the time after data deprivation. In the following, we used the taxonomy of SDM performance with AUC [47] The validation performances for the short-term future (2000–2015), i.e., up to 15 years after data deprivation, were *poor* (mean AUC = 0.64) but significantly better than random and comparable to the training performances (mean AUC = 0.58) (see Figure A10 of Section A.7). In other words, the predicted population in the presence cells was higher than the one of the non-detection cells in 64% of the random pairs of detection/non-detection validation cells. However, the predictions *failed* (average AUC = 0.44) in validation after 16 years of data deprivation (2016–2021), with performances worse than a random guess, while the training performance was *fair* (mean AUC = 0.73) in the same period. 

**Signals of introduction hotspots and residence time.** The estimated initial population sizes in 1980 varied significantly across the 24 introduction cells. Indeed, despite some uncertainty, the posterior estimates of the initial population size were significantly higher in some cells than others (Figure A5 of Section A.4). For instance, Figure 1—top shows that the highest population sizes (mean estimates) were in the north of Stellenbosch, near Paarl, Cape Town, and George. Nevertheless, significant populations were also located in other less urbanized areas. The oldest initial population was estimated to be in a coastal cell located around Wilderness, on the east side of George (Figure 1—bottom), where the mean age of the plants was nearly 40 years, suggesting the existence of (cultivated) plants in the area even before 1940, while the first BODATSA observation in the whole study area was recorded in 1963.

**An early massive spread wave driven by human-mediated long-distance dispersal.** The long-distance spread of seeds from urban areas happened consistently every year of the modeled period since 1980, with a higher impact on the invasion dynamics in the first 15 years. Indeed, while population sizes were low everywhere in 1980, by 1996 large populations had colonized the vicinity of two introduction areas, Stellenbosch and George, but also areas much further from the introduction sites, such as the vicinity of Swelledam in the middle of the study area (Figure 3). This was due to the long-distance dispersal from the urban areas of introduction that was already occurring in 1980 (top-left map of Figure 4). To prove that long-distance dispersal was necessary, we must compare what would have been the population dynamics without this dispersal mode. This counterfactual evidence is provided in Figure 7 (red curve)**,** showing that without long-distance dispersal, the total population would have been 10 times smaller in 1990, and still 4 times smaller by the end of the study period, despite the slow catching up due to short-distance dispersal. Additionally, the species would have been far from reaching its current equilibrium. The other alternative scenario (blue curve in Figure 7), where the model is deprived of short-distance dispersal while keeping all the other parameters constant (blue curve), showed that the absence of short-distance dispersal can hardly affect the invasion dynamics, except for a slight delay in overall population growth from 1990 to 1995.

**A fast establishment phase driven by local reproduction.** The total population grew by about a million times between 1980 and 1996 (Figure 5—top), leading to a steady state due to the self-regulation in our model (carrying capacity). The steep and sudden increase in the Shannon entropy (Figure 5—middle) shows that the population sizes were rapidly balanced between cells over the period of 1987–1994. This is explained by a synchronous growth in environmentally suitable cells driven by local dispersal. Indeed, it is crucial to highlight that long-distance dispersal was not intense enough to drive population growth by itself. It only allowed the establishment of small pioneer populations in many remote areas, while self-sustained local dispersal was responsible for driving their fast growth in a second phase. Indeed, long-distance dispersal resulted in far too few new plants annually ($10^{2}$ to $10^{6}$, Figure 6) to compensate for the overall annual mortality (50 +/− 20% of the total population, namely $10^{7}$ to $10^{9}$ annual deaths, according to Figure 5—top).

**The time before reproductive maturity induced marked growth steps.** As visible in Figure 8, representing the scaled fecundity curve, the age before reproductive maturity was estimated to be almost certainly three years. Indeed, despite a slightly earlier optima in the prior distribution of this parameter and under the model assumptions, the result suggested that the individuals were effectively reproductive in their third year. The fecundity increases quickly with age in older plants (Figure 8), although fecundity saturation is certainly not reached during the life span of most individuals, as 99% of plants die before the age of 4 to 13 years (given the uncertainty about mortality). Actually, the absolute fecundity only becomes greater than 100 with a probability of 0.95 at 5 years old, as illustrated by the wide confidence interval on fecundity between age 4 and 20 years (Figure 8) due to the uncertainty of the allometric scaling factor θ (estimated = 8.8 +/− 2). We thus conclude that only a very small proportion of germinated plants end up contributing significantly to the population growth in the fitted model. This latency phase before individual plants become significantly fecund also explains the two marked steps in the growth of the global seed production over time (Figure 5—bottom), and thus the lag phase. Note, we scaled the fecundity by its maximal value *M* in Figure 8. Indeed, because the model did not gain any information on this parameter from the data, its posterior distribution was the same as its prior.

**A possible lag phase of fifty years.** Our results showed that the species spread rapidly across the study domain, but the last population growth phase, which multiplied the population size by nearly 20, only occurred in the early 1990s (Figure 5—top). Given the inferred age-structure of the initial populations, which suggests an introduction prior to 1940, the model predicted a lag phase of 50 years or more preceding this last population growth phase. 

## 4. Discussion

We introduced a new Bayesian dynamic species distribution model to reconstruct the invasion dynamics of a perennial plant species from the occurrence data. This type of model merges the strengths of mechanistic dynamic models that may produce realistic and extrapolatable predictions, as shown here, with those of statistical Bayesian inference that leverages heterogeneous data and prior knowledge for parameter calibration. Although this large class of models has been used for related purposes [22,35,36,48], their potential to reconstruct and predict biological invasions has been little touched upon to date. In the model, we have accounted for two dispersal modes and age-structured fecundity during the transient phase of invasion dynamics. We exemplified it with the recent invasion of *Plectranthus barbatus* in South Africa. Our results highlighted the crucial importance of long-distance dispersal in determining the past spread and, indirectly, population growth (and invasiveness) of the species, most likely mediated by humans. Indeed, according to our estimates and ablation study, although *Plectranthus barbatus* reached an equilibrium population by 2000 or earlier, it would have been far below this equilibrium over the same period without long-distance dispersal. Hence, our results support the opinion that urban areas act as launching sites for invasive plants in natural areas of South Africa [49].

The fitted ecological process model could also be useful for formulating management strategies by testing the impact of different action scenarios. We are, however, not aware of any substantial management efforts directed at *Plectranthus barbatus* in South Africa in the past, which means that the invasion dynamics of the species have not been affected (negatively or positively) by management. This makes our model a coherent simulation tool for assessing what impact an eradication plan could have had on the invasion trajectory. More specifically, our complementary simulation, redrawing the invasion trajectory of the species without human-mediated long-distance dispersal, could actually be interpreted as a scenario of what the invasion would have been if the plant had been extirpated early and systematically around urban areas. Indeed, this simulation suggested that around 1996 the population would have been suppressed by 10 compared to the historical scenario where it had already reached equilibrium at this time. This strategy could buy time for broad-scale management and regulations to be coordinated. Whereas South Africa has advanced strategies for managing invasive species, none are systematically implemented for invasions in urban areas [50]. Urban-oriented management strategies are also less costly than actions in remote or protected areas. For instance, such strategies could more easily involve volunteers from citizen science programs by organizing mutually beneficial actions [51]. Our results on the age-structured fecundity also suggest that removing only the oldest plants (e.g., older than 5 years for *Plectranthus barbatus*), which is only a tiny proportion of the population but several orders of magnitude more fecund, would drastically reduce the management effort for the same level of control effectiveness.

A limitation of our study is that we did not account for introductions that may have happened in unreported areas or later than 1980, likely in the context of private gardens. This possibility is supported by the fact that our results suggested a first introduction prior to 1940 followed by a 50-year lag phase, which would have given more than enough time for nurseries and gardeners to propagate the species. The knowledge of early introductions in other sites would have likely decreased the estimated importance of long-distance dispersal in our model, but it would not alter our conclusion that humans have been crucial in propagating the species, ensuring its fast invasion dynamics. The intensity of gardening and the residence time have been shown to be major factors explaining the chances of cultivation escapes and the naturalization of woody plants [1]. However, this interdependent knowledge of planted population sizes, residence time, and lag phase is often lacking or approximative for non-native plant species [52]. Our model actually provided information on initial population size and age structure, filling these knowledge gaps.

If this study has revealed the hidden value of massive and heterogeneous crowdsourcing records (e.g., iNaturalist) for invasion monitoring, it also supports the broader call for better access to biodiversity data in urban areas in order to better manage the regional landscape [53]. For instance, improving the knowledge of urban plant ecology is tied to the availability of better data on urban biodiversity habitats [54]. In addition, large biodiversity occurrence datasets explicitly tagging cultivated (or captive) versus wild alien individuals, such as those hosted by iNaturalist, are rare. Such information is highly relevant for quantifying the propagule pressure during the naturalization phase, which is crucial for determining later invasion success and could be better accounted for in the kind of model proposed here. Another way to address this challenge of implicit tagging would be to apply automated procedures to classify a posteriori biodiversity records as cultivated/captive or wild.

We also showed that the last population burst of *Plectranthus barbatus*, which occurred in the early 1990s and led to the final equilibrium phase, was most likely preceded by at least 50 years of apparent lag phase since its introduction time, which is coherent with the lag phases of naturalized shrubs in the literature [8]. In agreement with our initial hypothesis, we showed that this lag phase is partly explained by the age-structured fecundity of *Plectranthus barbatus*. Indeed, according to our model reconstruction, the last decade of this lag phase, between 1980 and 1990, was due to the interplay between the low initial population size, the delayed reproductive maturity, and the high mortality. Empirical evidence for these types of phenomena, often documented in theory [29], have been scarce until now [26]. However, our model did not account for the period before 1980, when the data were too scarce, and the role of the age-structured fecundity in this early phase remains uncertain. In addition, even though our model inference accounted for the spatio-temporal variations of the sampling effort, the very low sampling effort in the early phase of the invasion questions the ability of our model to capture a significant signal of age-structure changes in the early populations and, hence, the large uncertainty of our reconstruction for this period. This brings us back to the harsh reality of the difficulty of separating a real invasion lag phase from the seemingly similar effect of an increasing sampling effort over time [55]. Still, our methodology should become more powerful given the increasing pace of museum data digitization [56] and of data collection through crowdsourcing.

Although our approach targeted interpretation and not future prediction, evaluating model fit and its future predictions through validation enabled us to characterize the relevance of the modeled processes in capturing the complexity of the drivers behind the actual invasion dynamics. The AUC performance on the training data was poor to fair (using the performance taxonomy from [47]) likely due to the relative simplicity of our environmental suitability model. The validation performances were poor in the short-term future, and with the training performances in the same period, but still significantly greater than random predictions and comparable to the AUCs obtained by [57], who evaluated SDM spatial transferability on 54 plant species. Hence, the model seems to have captured some transferable features for the environmental preferences, demographic, and dispersal rates of the species. However, the failure of long-term predictions in validation, after 16 years of data deprivation, illustrated the many sources of errors that may affect this kind of projection and their evaluation. Indeed, on the one hand, the predictability of invasions has often been fundamentally questioned [16,58], especially due to the intrinsic stochasticity of demographic events and inter-individual variability, and thus, it seems undeniable that predicting the future invasion dynamics remains an open problem. In addition, various other reasons can explain these overall low validation performances here, including the amount of false absences hidden in the non-detections used in the validation, and the many other potential environmental or biotic drivers not accounted for here.

The estimation precision of the current framework could be significantly improved by using more and/or better prior knowledge. We used informative prior distributions on the carrying capacity, which constrained the population size and the detection rates given the observed data and reduced the identifiability issues on these quantities [39]. Even though reduced, this identifiability issue was still present, as indicated by the residual correlation between the estimated parameters. Furthermore, the absence of evidence of information gain on the maximal fecundity from fitting the data questions the model design and its input data structure. Further work should determine whether the lack of parameter identifiability and information are structural or if information could be gained with more or better data structure and prior knowledge. Such questions could be addressed using simulations and Bayesian identifiability diagnostic tools [59]. Some iconic invasive species have a better documented physiology, ecology, and history of introduction and spread, providing the opportunity to use such greater knowledge to specifically reduce uncertainty on the population size and demographic rates. For instance, reproductive traits could be used to constrain more precisely the features of the age-structured fecundity curve, such as the maximum number of seeds per plant and the age before maturity, and account for vegetative reproduction. Estimates of detection probabilities obtained at finer spatial scales than the dense presence-only datasets and site-occupancy models [60] could inform prior distribution on the detection rates. Finally, the approximate age of reported individuals could be determined from images of crowdsourcing records (e.g., iNaturalist and Pl@ntNet) and could be added as a complementary data and fed into the model to better inform the estimation of mortality.

Although Bayesian inference is an important aspect of our methodology, given its capacity to account for prior knowledge on parameters, it induced algorithmic difficulties and heavy computational costs due to the complexity of our model. Indeed, we faced important convergence issues with our Metropolis–Hastings MCMC sampling algorithm. Initializing the algorithm with parameters of maximal posterior likelihood was crucial for obtaining a reasonable level of convergence. However, refining these parameter starting values required heavy preliminary runs. Moreover, the high-level sample auto-correlations in the final chains, due to an extremely low acceptance rate, required drastic thinning intervals, attesting to the inefficiency of the MCMC algorithm for exploring the parameter space (or the presence of a flat and rugged likelihood surface). Efficiently sampling complex and potentially multi-modal posterior likelihood surfaces is a known fundamental problem in Bayesian statistics [61], even though recent MCMC algorithms such as differential evolution MCMC [62] have been successfully applied to infer multi-species dynamic interaction models [48].

## Figures and Tables

**Figure 1 biology-11-01293-f001:**
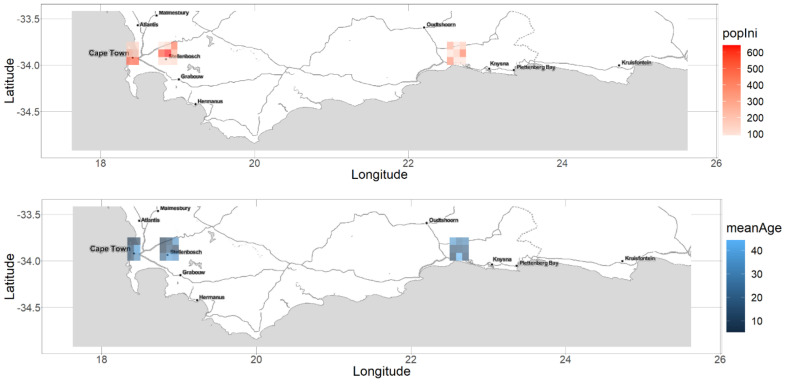
Maps of estimated mean population sizes and ages in the initial year 1980. **Top:** Mean posterior estimate of population size across the introduction cells. **Bottom:** Mean posterior estimate of the mean population age across the introduction cells. The associated posterior distributions are provided in Figure A5 and Figure A6 of Section A.4.

**Figure 2 biology-11-01293-f002:**
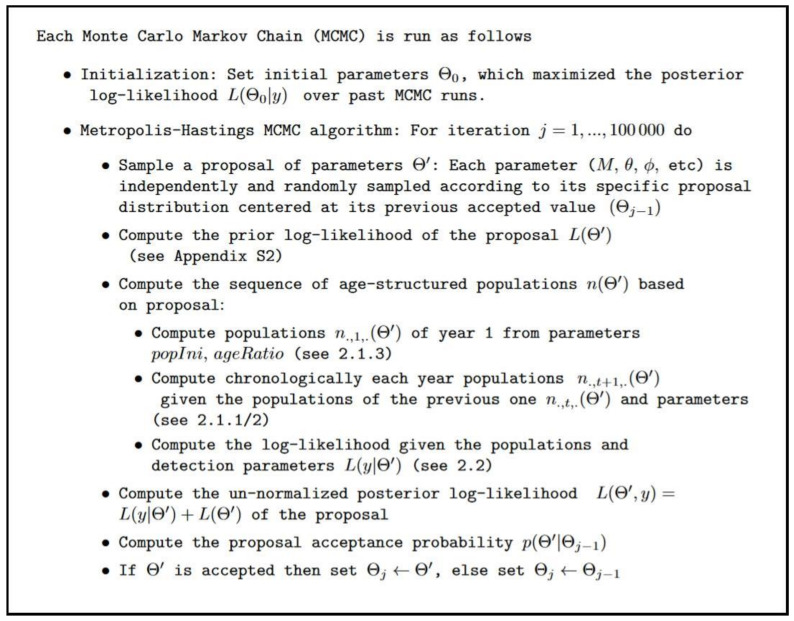
Summarized model fitting procedure for a single Monte Carlo Markov chain with Metropolis–Hastings sampling algorithm. The parameters $\Theta_{0}$ included 48 parameters for the initial populations, 14 for the ecological process, and 3 for the sampling process.

**Figure 3 biology-11-01293-f003:**
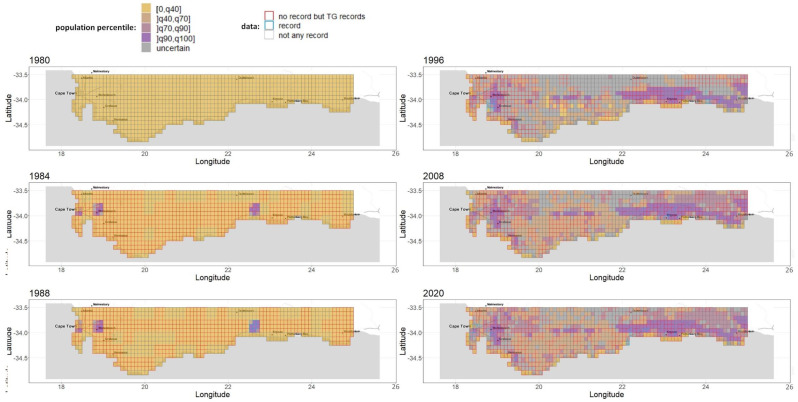
A reconstruction of *Plectranthus barbatus* invasion in the Southern Cape, South Africa, Part 1. Maps of population size percentile range for selected years between 1980 and 2021.

**Figure 4 biology-11-01293-f004:**
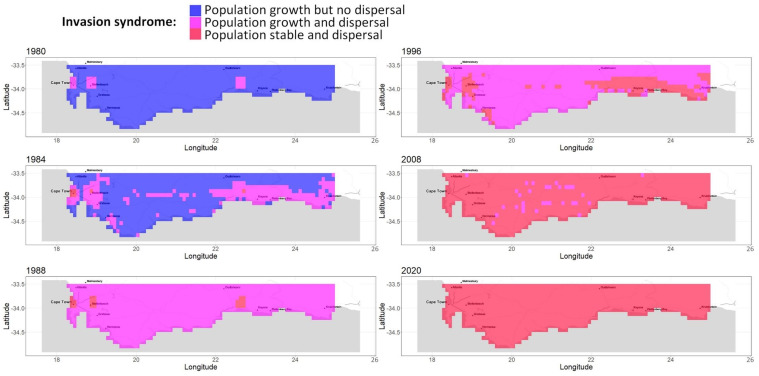
A reconstruction of *Plectranthus barbatus* invasion in the Southern Cape, South Africa, Part 1. Maps of population growth syndrome for selected years between 1980 and 2021.

**Figure 5 biology-11-01293-f005:**
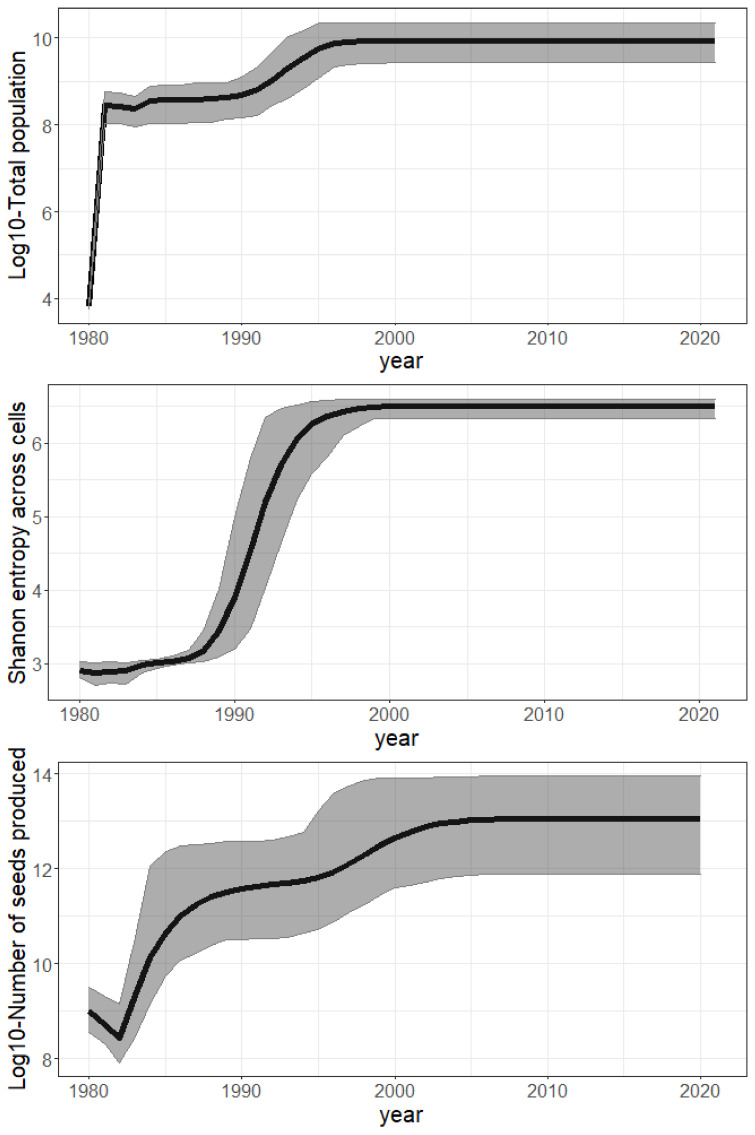
Global invasion metrics across sample parameters (mean / black line and 95% confidence interval / grey ribbon) over the study period (1980–2021). **Top:** Log10 of total population size per year. **Middle:** Number of 100 km² spatial cells colonized per year. **Bottom:** Log10 of total seed production per year.

**Figure 6 biology-11-01293-f006:**
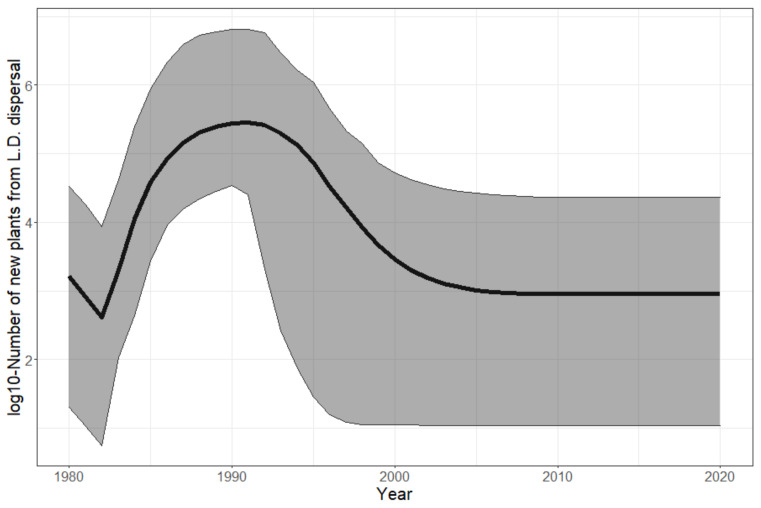
Contribution of long-distance (LD) dispersal to population recruitment over years. The posterior mean (solid curve) of the log10-number of new plants growing from seeds dispersed by long-distance dispersal is shown with its 95% confidence interval (grey ribbon) for each year.

**Figure 7 biology-11-01293-f007:**
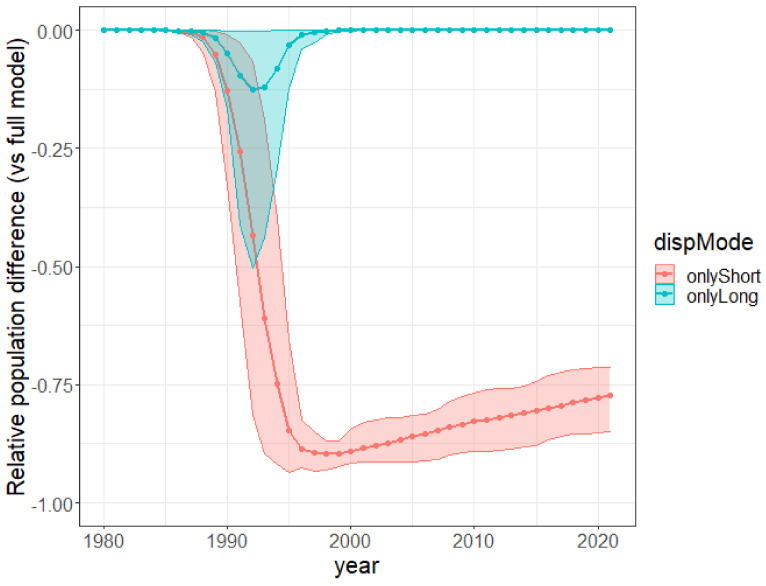
Relative reduction in population size under ablation of long-distance dispersal (red) or short-distance dispersal (blue) per year. For each ablated dispersal mode, we show the mean and 90% confidence interval (across posterior samples) of the population difference between the ablated model and the full one divided by the population of the full model.

**Figure 8 biology-11-01293-f008:**
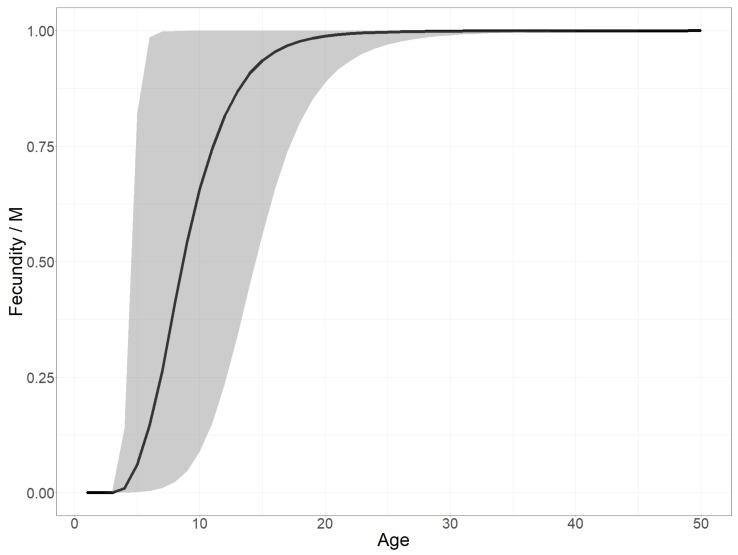
Posterior age-structured fecundity scaled by the maximal fecundity (*M*): mean posterior estimate (solid black line) and 95% confidence interval (gray ribbon). We first computed the curve from the three fecundity parameters ($\hat{k},\theta ,M$ for each posterior sample and then calculated the mean and quantile values (2.5% and 97.5%) per age.

## Data Availability

Data used in this study were collected from various sources mentioned in the Materials and Methods sections. The pre-formatted data openly provided for reproducibility are available from Zenodo (V3 https://doi.org/10.5281/zenodo.6921965, accessed on 20 August 2022). The R scripts (and pre-fitted parameter samples) to reproduce the last MCMC session, the convergence assessment, and all Figures are openly provided in the article’s GitHub repository: https://github.com/ChrisBotella/plectranthus_barbatus (accessed on 20 August 2022).

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
