# Peer review of "Dynamic Species Distribution Modeling Reveals the Pivotal Role of Human-Mediated Long-Distance Dispersal in Plant Invasion"

_biology, 2022, doi:10.3390/biology11091293_

Round 1

Reviewer 1 Report (Previous Reviewer 1)

I am very happy with the new version of that work. I found the text much clearer and the authors did a great job in improving the estimation process making the results stronger. They also acknowledge clearly the limits of the data and the model to avoid any over-interpretation of the results.

Looking forward to see this paper published!

Best regards.

Three little points:

* There is a problem with the display of equations certainly due to the submission process making the reading a bit difficult.

* L182-183: please add the parameters that correspond to each term of the product in order to define the parameters of equation 2.

* L205-207: Is that like reflecting boundaries? If yes you could add this here.

Author Response

Dear reviewer,

Thank you for going through this new version of our manuscript. We are pleased that it addressed your important concerns. We answer the new concerns point by point below.

"* There is a problem with the display of equations certainly due to the submission process making the reading a bit difficult."

-> Answer: Indeed, the probem of display of the equations, with exponents and subscripts not being displayed correctly, appears to be due to a change of format of our manuscript during the editorial process. In this newly revised version, we have checked and fixed all equations directly in the editorial version of the manuscript.

"* L182-183: please add the parameters that correspond to each term of the product in order to define the parameters of equation 2."

-> Answer: Equation 1 and 2 should now be displayed correctly and all their terms and corresponding parameters are explained in the text following them.

"* L205-207: Is that like reflecting boundaries? If yes you could add this here."

-> Answer: I am not sure it could interpreted as reflecting boundary. Long-distance dispersal could be understood as instantaneous and homogeneous diffusion of a small portion of seeds restricted to the study domain.

Best regards,

The authors

Reviewer 2 Report (Previous Reviewer 2)

No further comments 

Author Response

Thanks for going through our manuscript again.

Best regards,

The authors.

This manuscript is a resubmission of an earlier submission. The following is a list of the peer review reports and author responses from that submission.

Round 1

Reviewer 2 Report

General comments        

Overall the study is innovative as it tries to break the gaps between mechanistic model limitations when applying in data-driven aspects by combining both. Such an approach is innovative and the study goal bringing new knowledge in entropogenic effect on invasives plants dispersal. The model is strong and based on the species' bioecology to be modelled. The paper is well written, with every section is well explained. However, the introduction is too long and somehow confusing to clearly determine the present manuscript's target.

Specific comments

  1. The introduction too long and display unnecessary information. Kindly focus on the target goal and only indicate how your work will help achieve your final goal.
  2. This paper is a methodological paper that provides a self-explanatory conceptual framework to strengthen this section.
  3. The model validations results did not come up clear expecially in the abstract, Kindly provide others metrics to indicate the accuracy of the models. Also, is 65% a very robust accuracy? If yes provide evidence (cite a reference(s))

Decision

This paper should be accepted upon revision and addressing the above comments.